# Adaptation and Validation of the Individual Lifestyle Profile Scale of Portuguese Older Adults Living at Home

**DOI:** 10.3390/ijerph19095435

**Published:** 2022-04-29

**Authors:** Ana da Conceição Alves Faria, Maria Manuela Martins, Olga Maria Pimenta Lopes Ribeiro, João Miguel Almeida Ventura-Silva, Paulo João Figueiredo Cabral Teles, José Alberto Laredo-Aguilera

**Affiliations:** 1Abel Salazar Biomedical Sciences Institute, University of Porto, Rua Jorge de Viterbo Ferreira, 228, 4050-313 Porto, Portugal; anacafaria85@gmail.com (A.d.C.A.F.); enf.joao.ventura@gmail.com (J.M.A.V.-S.); 2North Region Health Administration, 4000-447 Porto, Portugal; 3Nursing School of Porto, 4200-072 Porto, Portugal; mmartins@esenf.pt (M.M.M.); olgaribeiro@esenf.pt (O.M.P.L.R.); 4Center for Health Technology and Services Research, 4200-450 Porto, Portugal; 5Centro Hospitalar, Universitário de São João, 4200-319 Porto, Portugal; 6School of Economics, University of Porto, 4200-465 Porto, Portugal; pteles@fep.up.pt; 7Laboratory of Artificial Intelligence and Decision Support-INESC Porto LA, 4200-465 Porto, Portugal; 8Facultad de Fisioterapia y Enfermería, Campus de Fábrica de Armas, Universidad de Castilla-La Mancha, Av de Carlos III, nº 21, 45071 Toledo, Spain; 9Multidisciplinary Research Group in Care (IMCU), University of Castilla-La Mancha, 45005 Toledo, Spain

**Keywords:** validation studies, factor analyses, aged, lifestyle

## Abstract

(1) Background: Unadjusted lifestyles have been the main cause of risk for the loss of years of healthy life. However, currently valid and reliable instruments to assess the lifestyles of the elderly are quite long and difficult to interpret. For this reason, the objective of this study was to adapt and validate the ‘Individual Lifestyle Profile’ (ILP) scale in a sample of elderly people; (2) Methods: A methodological study was carried out and a sample of 300 older adults enrolled in a Health Unit located in the North of Portugal was used, who responded to the scale. We examined internal consistency, predictive validity, and discriminative ability; (3) Results: After the Exploratory Factorial analysis, a solution was found with four factors that explain a variance of 67.8%. The designation of the factors was changed from the original scale, with the exception of one dimension, and they were called Health Self-management, Social Participation and Group Interaction, Citizenship and Physical Activity. The total internal consistency (Cronbach’s alpha) was 0.858, ranging from 0.666 to 0.860 in the mentioned factors; (4) Conclusions: The ILP scale proved to be easy to apply and presented a good reliability and validity index, based on internal consistency, AFE and AFC. The scale allows evaluating the lifestyle of older adults, and its use will be aimed at modifying behaviors associated with negative lifestyles of older adults and their individual needs.

## 1. Introduction

Population ageing represents one of today’s main challenges. Globally, it is estimated that the number of people over 60 will triple in the next 30 years [1]. In the European Union, the proportion of people aged 80 and over is expected to increase two and a half times between 2019 and 2100 [2].

However, although we recognize that the increase in average life expectancy is an important achievement [3], we highlight the fact that in industrialized countries, alcohol abuse, smoking, sedentary lifestyle, and inadequate diet remain the main risk factors for the loss of healthy life years [4,5]. We know that the most important factors determining health in society are lifestyles and the associated health behaviors, in addition to socioeconomic, cultural, and environmental conditions, as well as the community support network [6]. Ageing, unhealthy lifestyles, and negative socioeconomic and environmental conditions lead to an increase in the prevalence of chronic diseases and a reduction in the functional capacity of the older adults, which limits their independence, autonomy, and social participation [3,4,5,6]. In 2015, a world report on aging and health was published in which the term intrinsic capacity was introduced [7]. This term is composed of individual physical and mental capacities, focusing on capacities (rather than deficits) based on body functions for healthy ageing and their interactions with environmental characteristics [8]. Therefore, the intrinsic capacity could discriminate older adults depending on their risk of care dependency, with the possibility of early detection of decreases or low levels of it. [9]. This evaluation of biological age would help to understand the functional evolution and vulnerabilities of older adults, and to be able to act preventively or treat early to avoid loss of autonomy [10,11].

It is of primordial importance that people are proactive in managing their own health and consciously make healthy choices such as following a healthy diet, controlling weight, engaging in regular physical activity, having adequate sleep hygiene, managing stress, refraining from smoking and alcohol consumption, and immunizing themselves against diseases through vaccination [12,13].

As there seems to be a relationship between ageing, lifestyles and chronic diseases, assessing older adults’ attitudes and behaviors towards a particular lifestyle can furnish relevant information for health professionals to plan interventions that increase older adults’ literacy and sustained decision-making, hence improving their health [14].

Thus, the Directorate-General of Health (DGS) and the World Health Organization (WHO) underline the importance of health professionals who exercise their functions in the community in implementing programs that promote health literacy, healthy lifestyles, and empowering self-management to cope with disease processes, hence delaying the decline in functional capacity and frailty that occurs with ageing [4,15].

After conducting the literature review, we found some instruments available that help assess lifestyles that promote healthy ageing. However, they are mostly long, self-filling, and difficult for people with lower levels of education to interpret [16,17]. The most commonly used are the Health-Promoting Lifestyle Profile (HPLP) Questionnaire [18] and the Health Enhancement Lifestyle Profile (Enhancement Lifestyle Profile (HELP) Questionnaire [19]. The Fantastic Lifestyle Questionnaire [20], the Breslow’s Lifestyle Index Questionnaire [21], and the Individual Lifestyle Profile Questionnaire [22] with 15 items, not yet validated for the Portuguese population, are also used.

In this sense, the objective of the present study is to adapt the European Portuguese context and describe the psychometric properties of the ‘Individual Lifestyle Profile’ (IHLP) scale version in a sample of older adults living at home.

## 2. Materials and Methods

Given the objectives of the study, we carried out a methodological study using a quantitative research approach for the adaptation and validation of instruments [23]. We adopted, as theoretical reference, the model suggested by Pasquali to assess the validity and reliability of the questionnaire, which is through psychometric analysis [24].

### 2.1. Participants

A cross-sectional design was used in this methodological study. A non-probabilistic sample was used and consisted of 340 older adults, which is adequate considering the recommendation set by COnsensus-based Standards for the selection of health Measurement INstruments (COSMIN), which suggests six to ten observations for each item of the instrument that need to be validated, and having no less than 50 participants [25]. Inclusion criteria were: being 65 years of age or older, enrolled in a health care facility in Northern Portugal, and have preserved cognitive ability ascertained at the beginning of the interview, supported by the following items: orientation and registration memory of the Mini-Mental State Examination Scale (MMSE) [26]. The exclusion criteria were: having a communication impairment, total dependence in self-care, and being institutionalized.

### 2.2. Data Collection and Procedures

The identification and selection of the older adults was carried out by professionals of Health Unity who, following the list of the 2300 older adults enrolled and, according to the inclusion and exclusion criteria, proceeded with the recruitment process by telephone contact. After the older adults had accepted to participate in the study, the interview dates were scheduled according to the research team (the main researcher and three nurses previously trained to apply the form and) and the older adults’ availability.

The data collection took place between October 2020 and May 2021, and the average application time for each interview was 20 min.

During the recruitment, 40 older adults refused to participate and 300 older adults accepted to participate.

### 2.3. Instruments

The researchers designed the form that comprises the sociodemographic and health characterization of the older adults (sex, age, marital status, education, cohabitants, pathological background), and the ‘Individual Lifestyle Profile’ Scale [24]. This scale evaluates people’s lifestyles, based on the Well-Being Pentacle model. The original version was conceived in Brazil by Nahas, Barros and Francallini (2000), and validated by Both et al. (2008) with reliability (Cronbach’s alpha) of 0.78 and a variance of 58.65%. After we requested the authors, they gave permission to use and validate the ILP. The scale includes 15 questions, subdivided into five components namely, nutrition; physical activity; preventive behavior; relational behavior, and stress control. In each component, three questions are placed and the response options are given on a *Likert* scale, ranging from 0 (never), 1 (sometimes), 2 (almost always), and 3 (always). Answers 0 and 1 indicate health risk behavior (negative profile) and answers 2 or 3 are positive indicators. In each component, the interpretation follows the same logic, nonetheless, it is suggested to classify the sum of the three items of each of the five components as follows: up to 3—negative profile; 4 to 6—intermediate (can improve), and 7 to 9—positive profile. The lower the score obtained, the greater the need for behavioral change.

For the linguistic adaptation and validation of the scale to European Portuguese, the content of the items was validated. Although the scale was developed and presented in Portuguese, there was a terminology specific to Brazil. As such, the scale was analyzed by five judges, which led to the modification of seven items of the scale and a semantic analysis was carried out in order to analyze the understanding of the items. A pre-test was carried out with 48 older adults and all the changes presented were incorporated into the final version of the scale.

### 2.4. Data Analysis

For data analysis and processing, we used The Statistical Package for the Social Sciences (SPSS), version 26.0 (Armonk, New York, NY, USA).

Prior to the exploratory factor analysis (EFA) and to the confirmatory factor analysis (CFA), we analyzed the homogeneity of the items, namely the corrected inter-item correlation and the corrected total item correlation; consistency analysis, by calculating the reliability coefficients, namely Cronbach’s alpha coefficient (α); Guttman and Spearman Brown coefficients, and the calculation of the sampling adequacy through the Kaiser–Meyer–Olkin (KMO) index, whose required score should be greater than or equal to 0.60 [27].

We further assessed structural validity by using exploratory factor analysis (EFA) and confirmatory factor analysis (CFA).

We carried out factorial retention by using the principal components method, i.e., the Kaiser criterion, followed by varimax rotation. For factor retention, we took into account eigenvalues greater than one and the scree plot slope. As item saturation criterion, we considered values equal to or greater than 0.20 [23]. Subsequently, we checked the communalities and performed the Spearman’s correlation matrix between the subscales.

In the CFA, the quality of fit of the model was assessed, namely through the calculation of quality of fit coefficients such as the Comparative Fit Index (CFI), the Goodness-of-Fit Index (GFI), the Adjusted Goodness-of-Fit Index (AGFI), the Incremental Fit Index (IFI), the Tucker–Lewis Index (TLI), the Root Mean Square Residual (RMSR), the Root Mean Square Error of Approximation (RMSEA), and the Modified Expected Cross-Validation Index (MECVI).

We used Cronbach’s alpha coefficient (α) and composite reliability to estimate the reliability of the ILP scale [28]. We assessed factor validity by calculating the standardized regression weights of each item on the subscales and individual item reliability. We also assessed the measured convergent validity through the average variance extracted and the discriminant validity by determining the square matrix of the correlations of the subscales.

### 2.5. Ethical Considerations

The study was approved by the Ethics Committee of the health care institution where we conducted the research which gave us their approval (opinion number 24/2020), and we obtained the participants’ consent. We guaranteed anonymity and confidentiality of the data collection. All ethical and legal principles were met.

## 3. Results

Of the total 300 older adults who agreed to participate in the study, the majority were female (60.33%), married (58%), living only with their spouse (50.7%), and had 4 years of schooling (49.7%). The average age of the participants was 81.34 ± 6.75 years. Regarding the clinical characteristics of the older adults, the majority declared having some disease (98.3%), of which 78.7% reported having musculoskeletal and osteoarticular disease, 77.4% hypertension, and 31.7% diabetes mellitus.

With regard to the results derived from the application of the ILP scale, we firstly checked whether the data were appropriate to perform the factor analysis. We found that the inter-item correlations ranged between 0.015 and 0.753 (mean correlation of 0.3) with many moderate and high values, and the corrected item-total correlations ranged between 0.35 and 0.612 (mean correlation of 0.505), therefore, the homogeneity of the items constituting the scale is acceptable (Table 1).

Subsequently, we divided the scale into two parts to check the reliability coefficients (Cronbach’s alpha coefficients).

We then analyzed the reliability of the scale using Cronbach’s alpha coefficient, the item-total correlation, the inter-item correlation, the correlation between the two halves, Guttman’s split-half reliability coefficient and Spearman–Brown’s coefficient.

With regard to Cronbach’s alpha, by dividing the scale into two parts with a number of items as close as possible, namely eight and seven items, we considered the values moderate, 0.803 and 0.751, respectively.

We considered the average corrected item-total correlation and the average inter-item correlation acceptable, 0.505 and 0.300, respectively, since there were a significant number of moderate correlations between the two halves (0.650), which demonstrated convergent validity. The Guttman’s split-half reliability coefficient was 0.787 and the Spearman–Brown coefficient 0.788; they are both high, which demonstrates that the consistency of the scale is good.

The sample adequacy criterion was confirmed using the Kaiser–Meyer–Olkin (KMO) suitability test, and both the overall value (0.847) and the values for each item are above 0.7. Most of them are above 0.8. Hence, we can say that the factorability of the correlation matrix is good, and it is appropriate to perform a factor analysis with these data [28]. In this way, a factor analysis was performed with extraction of factors by using the principal components method, that is, after establishing the Kaiser criterion, which consists of selecting the factors whose associated eigenvalues are greater than 1, we reached a four-factor solution, explaining 67.8% of the total variance.

The second rule is to reconstitute 80% of the total variance, i.e., Pearson’s rule, which leads us to a seven-factor solution, explaining 81.7% of the total variance; however, it is too high and therefore, it is not suitable. The third rule is based on the ‘scree plot’, where we retained the number of factors in which the largest percentage drop in the variance explained occurred (Cattell’s rule), hence pointing to a four-factor solution.

After careful consideration, we adopted the four-factor solution, as it is the best in terms of interpretation and meaning of the factors, and explains an acceptable percentage of the total variance (67.8%) distributed in the following proportions, Factor ‘1’: 35.39%, Factor ‘2’: 16.24%, Factor ‘3’: 9.09%, Factor ‘4’: 7.07% (Table 2).

In order to ease the interpretation of the results of the factor analysis forced to four factors, the items are displayed according to the factor in which they saturated rather than in the order of the original scale. Thus, the item loadings in each factor are shown and the largest loading is highlighted. The factor eigenvalues and the respective percentages of the total explained variance are also displayed.

Both the factorial weights and the communalities are acceptable or high. A single communality is below 50%, being very close (44.6%).

After deriving the four-factor solution, we named the factors taking the theoretical framework into account, i.e., the ILP dimensions were named as: Health Self-management (F1), consisting of six items; Social participation and group interaction (F2), consisting of four items; Citizenship (F3) consisting of three items, and Physical activity (F4) consisting of two items.

Table 3 shows the distribution of the answers to the subscales (dimensions) identified, showing that the majority refers to not practicing physical activity (82.8%), nor participating socially or in groups (53.6%). Concerning the Health Self-management dimension, nearly 36.3%, always carry out activities within this scope, and only 12.8% say that it is not part of their daily routine. With regard to Citizenship, 35.4% say this dimension is always true.

The correlation matrix (of Spearman) between the subscales revealed significant and positive correlations between weak to moderate (from 0.179 to 0.457), and once more, it reveals that the outcomes of this factor analysis are of good quality.

We tested the tetra factor solution of this scale through confirmatory factor analysis.

Among other aspects, we analyzed the quality of model adjustment, scale reliability, factor, convergent, and discriminant validity.

Regarding the quality of the adjustment of the proposed model, we found that there are only 34 non-redundant residuals (i.e., 34%) with an absolute value greater than 0.05, which indicates a very good adjustment. Furthermore, the Comparative Fit Index (CFI) is 0.906, indicating an acceptable fit, the Goodness-of-Fit Index (GFI) and the Adjusted Goodness-of-Fit Index (AGFI) are 0.893 and 0.848, respectively, and therefore, we can consider that these values indicate a good fit. The Incremental Fit Index (IFI) is 0.907, which means it has a nearly good fit. The Incremental Fit Index (IFI) is 0.907, which means it has a good fit. The Tucker–Lewis Index (TLI) is 0.882, hence very close to being adequate, the Root-Mean-Squared Residual (RMSR) is 0.068, also considered as a good fit and the Root-Mean-Squared Error of Approximation (RMSEA) is 0.087, a value considered as acceptable. Finally, for comparison with other models, the Modified Expected Cross-Validation Index (MECVI) was also calculated and assumed a value of 1.17. All these coefficients indicate an acceptable or good fit.

Subsequently, to assess the reliability of the scale used, i.e., its internal consistency, Cronbach’s alpha coefficient and composite reliability were calculated.

The Alpha value for the total scale is 0.858, which is high and shows a good internal consistency of the scale. In addition, the consistency of the four subscales was 0.831, 0.860, 0.666, and 0.824, respectively, which reveals good consistency, except for the third subscale, whose consistency is acceptable.

The composite reliability of the four subscales was 0.851, 0.872, 0.679, and 0.828; hence, it is very high in all the subscales except for the third one, whose composite reliability is almost adequate. In this sense, it is possible to evaluate that both the overall scale and the subscales identified, generally reveal good or at least acceptable reliability and internal consistency in a subscale.

To assess factor validity, we calculated the standardized regression weights of each item in the various dimensions and the square of these weights, i.e., the individual reliability of the items according to Table 4.

With a single exception, all other items showed individual reliability higher than 0.25 (therefore, appropriate), however, most of them are much higher, hence we can affirm that all subscales present factor validity.

We measured the convergent validity of a subscale through its mean extracted variance (MEV) [23], and we found that the MEV of the subscales Health Self-management dimension, Social participation and group interaction dimension, Citizenship dimension, and Physical activity dimension were 0.493, 0.633, 0.431, and 0.707, respectively. The 2nd and 4th dimensions have the MEV clearly higher than 0.5, and the 1st dimension was approximately equal to this value, thus these subscales have adequate convergent validity. Only the 3rd dimension has the MEV slightly below 0.5, therefore its convergent validity is considered almost adequate.

With regard to discriminant validity, this occurs when the MEV of two subscales are greater than or equal to the square of the correlation between them [18], and it was found that the MEV of all subscales is clearly greater than the square of the respective correlation coefficient. Consequently, all pairs of subscales have discriminant validity, and it is therefore possible to affirm that the discriminant validity of the scale is good.

The factor model identified shows good quality, reliability, and validity, thus it is possible to affirm that it is appropriate.

## 4. Discussion

Identifying the lifestyles of the Portuguese older adults’ population is an increasingly national strategy to improve their quality of life. Thus, instruments that make it possible to assess lifestyles are essential for health professionals to define individualized strategies to promote healthy ageing. However, most validated instruments are long and difficult to apply in primary health care consultations and, in this sense, this study was conducted to adapt and validate to the European Portuguese context Nahas’ individual lifestyle profile scale, Barros and Francallini (2000), validated by Both et al. (2008), and based on the Well-Being Pentacle [24,29].

We distributed the values obtained from the scale across the intervals of the response scale; however, the answer ‘not part of my lifestyle’ was the most frequent answer on the global scale.

The results also demonstrate that the scale applied in Portuguese older adults presents reliability and validity, analyzing Cronbach’s alpha coefficient (α), the composite reliability, and factorial, convergent, and discriminant validity [28]. The KMO value obtained was 0.847, which is considered very good, evidencing that the items of the proposed scale measure the same construct and are inter-related, and it was found that the data matrix is capable of factoring [27].

The exploratory factor analysis with varimax rotation indicated the existence of a tetra factor construct, with the four-factor retained explaining 67.8% of the total variance.

The first factor showed high factor weights of the items included in the nutrition, preventive behavior, and stress control behaviors of the original version; hence, we decided to rename this dimension as Health Self-management. It is interesting to see that these three types of behaviors saturate in the same factor; this means that older people associate these behaviors mentally, hence reflecting this in their responses. People consider that nutritional care, control of analytical values such as blood pressure and cholesterol, not smoking or consuming alcohol (or consuming in moderation), and taking time to relax are part of the individual behaviors and attitudes promoting their health which they consider they should be able to self-control and self-manage. Pender (2015) mentions that health accountability is related to self-care for health promotion, and it requires that a person has health literacy and empowerment to maintain or improve their health. Therefore, it is necessary for people to be involved, aware, and feel responsible to play an important role as a main actor in the promotion and self-management of their health in terms of seeking information and making sustained decisions [12,30]. However, it is not only necessary for nurses to transmit adequate data, but above all, to have knowledge on the barriers to action, the feelings inherent to behavior, the perceived benefits, as well as the interpersonal and situational influences, so that they can provide the conditions to adopt healthy behaviors [31,32].

The second factor showed high factor weights of the three items of relational behavior and one of physical activity-related behavior from the original version. We found that these two types of behaviors saturated in the same factor, hence we suppose that older people may associate to them, and they may not have the habit of walking or cycling as a means of transport, nor do they frequently use the stairs in their daily lives. On the contrary, they should only do it when taking a walk and not as a means of transport, and when they do, it ends up being a social activity and interaction with other people rather than a predominantly physical activity. Although walking rather than using the car for short distances and using the stairs rather than lifts can be as effective as a structured exercise program [33], older people have not yet evidenced this benefit. In this sense, we named this dimension ‘Social participation and group interaction’ because it is essential in people’s lives. This dimension is in line with what we find described in several studies that report social participation and group interaction as factors promoting healthy ageing, since emotional support can help minimize stress and increase resilience when facing functional and cognitive decline that occurs with ageing [34,35]. Sharing experiences stimulates a sense of meaning or coherence in life, as well as positively influencing health behaviors [34]. They also help to share social and family concerns, beliefs, fears, difficulties, and moments of crisis, as well as to avoid loneliness and isolation [12].

The third factor presented high factorial weights in one preventive behavior item and two stress control items in the original version and, consequently, we called it ‘Citizenship’. The surveyed older adults associate preventive behavior with ethics, citizenship and well-being at a personal and societal level. Indeed, the exercise of citizenship, and active and thoughtful participation in society not only promotes health, but also the general well-being of the elderly [36,37].

It has already been defined in the Portuguese National Health Plan 2012–2016 that health citizenship is one of the four strategic axes, involving concepts such as behavior and lifestyles, health literacy, and chronic disease management, in a culture of commitment, self-control, responsibility, autonomy, proactivity, and active participation of the person [38]. Recently, through the National Strategy for Active and Healthy Aging 2017–2025, the Portuguese government committed itself to an action plan for the Portuguese elderly, which, in addition to prioritizing health, healthy lifestyles, health surveillance, and management of morbidity processes, also recommends promoting the exercise of citizenship [4].

Finally, the fourth factor showed high factor weights in two items of the physical activity behavior of the original version. In this dimension, we did not change the denomination ‘physical activity’. Since these two behaviors saturate in the same factor, it demonstrates that older people associate them to the need of regular participation, as defined by the author of the original version of the ILP and as defined by the WHO [33].

We strongly recommend light, moderate, and/or vigorous physical activity since its benefits are able to delay functional decline, prevent or control chronic diseases at both psychosocial and cognitive levels [33]. Pender (2015), in his health promotion model [12], highlights physical activity as a primary behavior for health promotion, as does Nahas et al. [24]. However, adherence to physical activity depends not only on the person’s individual characteristics such as motivation, self-efficacy, and motor skills, but also on environmental characteristics, which the nurses should be aware of in order to encourage this health-promoting behavior and arrange individualized strategies that meet the needs and profile of each elderly person [39].

We defined all of the dimensions according to theoretical foundations and bibliographical research that support the process of adoption of lifestyles, and hence grouped the 15 items of the ILP in the already mentioned four dimensions and expressed in Figure 1.

Regarding the communalities, these were acceptable or high, with the exception of one item, 13, that presented slightly 44.6%. Additionally, when we assessed the individual reliability of this item, we found it appropriate (0.305), and therefore, we kept the item in the final version.

With regard to the AFC, its objective was to confirm and adjust the theoretical model proposed for the scale [40], and the analysis revealed a good, or at least acceptable adjustment. Reliability and internal consistency of the instrument were good, or at least acceptable. We found that the total Cronbach’s alpha coefficient obtained was 0.858. It was higher than the value obtained by Both et al. (2008), 0.78, when applying the ILP scale to a physical education teachers’ sample and it was also higher than that obtained by Hernandez et al. (2007), 0.71, in a sample of 168 adults aged between 30 and 68 years [41].

In our study, Cronbach’s alpha values of the four dimensions were reasonable and good, ranging from 0.666 to 0.860. While in the study by Both et al. (2008), Cronbach’s alpha values in the five dimensions ranged between 0.48 and 0.67, i.e., weaker values when compared to our research [29]. 

Composite reliability was also very high in all of the dimensions except in the Citizenship dimension, which was almost adequate. In addition, the factor, convergent, and discriminant validity of the scale were good. Another indication that supports the validity of the ILP scale is the fact that it shows statistically significant correlations between the new structure of the dimensions, and therefore, they are not redundant, which leads us to assess the different aspects of the same construct and to conclude that the four-factor model presented is appropriate [40].

### Limitations and Strength

A limitation of this study is the fact that it was applied only in one Health Unit in the Portuguese context, that is, to a sample with a specific sociocultural context. New studies should be carried out with larger samples and with different characteristics to reinforce the validity and reliability in older adults from other geographic areas, as well as explore its association with other variables.The greatest strength of this study is that it determined the validity and reliability of this scale in the European Portuguese context. The process of cultural adaptation and validation followed the quality procedures defined in international consensus, resulting in this version of the assessment of the lifestyles of the older adults who live at home in primary health care settings.

## 5. Conclusions

The ILP scale proved to be easy to apply, and presented a good reliability and validity index, based on internal consistency, AFE and AFC. The psychometric adequacy of the ILP scale for the Portuguese population indicates that it can be used in future studies with the objective of analyzing the lifestyles of the older adults. It has proved to be a valuable tool to support the decision-making of nurses and other health professionals in primary care services in health promotion aimed at modifying behaviors associated with the negative lifestyles of the older adults and their individual needs.

The adoption of healthy lifestyles and active participation are the most important self-care throughout the life cycle, and as such, it is very important that an early intervention is carried out in the face of behaviors and lifestyles that increase risk development of pathologies, hence the relevance of this scale and its great utility in future studies both in clinical practice and in research.

Early intervention and the promotion of awareness of the need to change behaviors in the face of negative lifestyles will allow the older adults to maintain a better quality of life for as long as possible.

## Figures and Tables

**Figure 1 ijerph-19-05435-f001:**
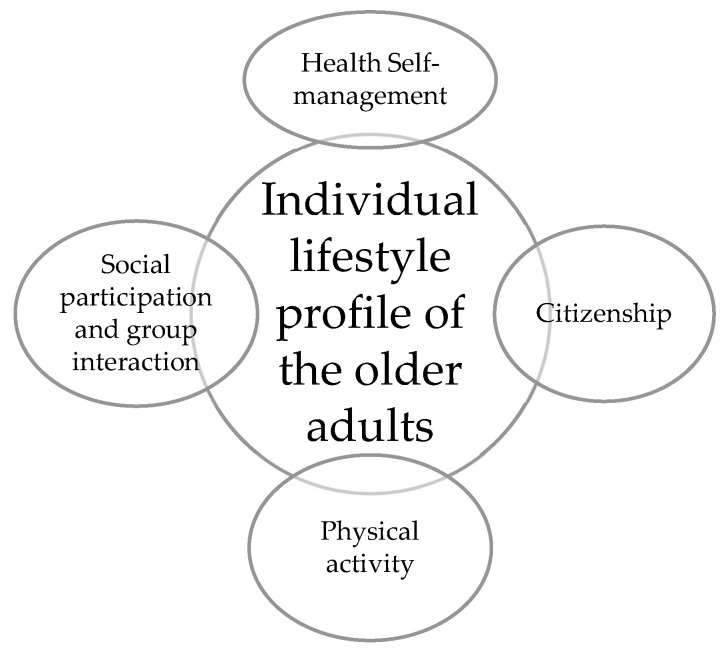
Dimensions of the individual lifestyle profile of the older adults that were corrected in the new framework (N = 300).

**Table 1 ijerph-19-05435-t001:** Corrected item-total correlations (N = 300).

Item	Correlation	Item	Correlation	Item	Correlation
1	0.594	6	0.612	11	0.540
2	0.580	7	0.569	12	0.580
3	0.508	8	0.413	13	0.471
4	0.471	9	0.350	14	0.405
5	0.365	10	0.539	15	0.582

**Table 2 ijerph-19-05435-t002:** Individual Lifestyle Profile Scale—factorial structure (N = 300).

Item	Factor 1	Factor 2	Factor 3	Factor 4	Communalities
1—Your daily diet includes at least 5 portions of fruit and vegetables	**0.780**	0.225	0.106	0.026	0.671
2—You avoid eating high fat food (fatty meats, fried food) and sugar	**0.820**	0.162	0.110	−0.021	0.711
3—On a daily basis you eat, 4 to 5 varied meals, including a full breakfast	**0.832**	0.175	−0.108	0.051	0.737
7—You know and control your blood pressure and cholesterol levels	**0.580**	0.155	0.385	0.072	0.514
8—You do not smoke or consume alcohol (or consume in a moderate way)	**0.723**	−0.091	0.157	0.029	0.557
13—You take a little time each day (at least 5 min) to relax	**0.630**	0.075	0.166	0.124	0.446
6—On a daily basis, the means of transport you choose is to walk or cycle and you prefer to use the stairs rather than the lift	0.235	**0.570**	0.339	0.283	0.575
10—You try to make friends and are satisfied with your relationships	0.158	**0.854**	0.105	0.032	0.766
11—Your leisure time includes meeting with friends, team sports activities, participating in associations or social entities	0.064	**0.831**	0.091	0.303	0.794
12—You are an active member of the community and feel useful in your social environment	0.137	**0.867**	0.092	0.209	0.823
9—You respect traffic rules (as a pedestrian, cyclist or driver) and if you drive, you always wear your seat belt and never consume alcohol	0.000	0.140	**0.738**	0.131	0.581
14—You keep up a discussion without losing your temper, even when contradicted	0.284	−0.040	**0.760**	−0.037	0.661
15—You balance time between work and leisure	0.171	0.368	**0.676**	0.146	0.643
4—You perform at least 30 min of moderate/intense physical activity, continuously or cumulatively, 5 or more days during the week	0.126	0.256	0.139	**0.863**	0.845
5—You do exercises involving muscle strengthening and stretching at least twice a week	0.015	0.238	0.066	**0.885**	0.844
V.P.	5.308	2.436	1.363	1.060	
% var.	35.385	16.238	9.089	7.065	

**Table 3 ijerph-19-05435-t003:** Individual Lifestyle Profile Scale—Distribution of the answers of the new structure in the dimensions (N = 300).

Dimensions	Does Not Make Part of		Sometimes		Almost Always		It Is Always True	
	*n*	%	*n*	%	*n*	%	*n*	%
1	231	12.8	475	26.4	654	36.3	440	24.4
2	643	53.6	326	27.2	155	12.9	76	6.3
3	160	17.8	204	22.7	217	24.1	319	35.4
4	497	82.8	72	12.0	20	3.3	11	1.8
Scale	1531	34.0	1077	23.9	1046	23.2	846	18.8

**Table 4 ijerph-19-05435-t004:** Individual Lifestyle Profile Scale—Standardized regression weights and individual reliability in the subscales of the new structure of the dimensions (N = 300).

Subscale 1—Health Self-Management Dimension	Regression Weight	Individual Reliability
Your daily diet includes at least 5 portions of fruit and vegetables	0.793	0.629
You avoid eating high fat food (fatty meats, fried food) and sugar	0.822	0.676
On a daily basis you eat 4 to 5 varied meals, including a full breakfast	0.775	0.601
You know and control your blood pressure and cholesterol levels	0.610	0.372
You do not smoke or consume alcohol (or consume in a moderate way)	0.615	0.378
You take a little time each day (at least 5 min) to relax	0.552	0.305
**Subscale 2—Social participation and group interaction** **dimension**	**Regression weight**	**Individual Reliability**
On a daily basis, the means of transport you choose is to walk or cycle and you prefer to use the stairs rather than the lift	0.647	0.419
You try to make friends and are satisfied with your relationships	0.778	0.605
Your leisure time includes meeting with friends, team sports activities, participating in associations or social entities	0.847	0.717
You are an active member of the community and feel useful in your social environment	0.889	0.790
**Subscale 3—Citizenship dimension**	**Regression weight**	**Individual Reliability**
You respect traffic rules (as a pedestrian, cyclist or driver) and if you drive, you always wear your seat belt and never consume alcohol	0.464	0.215
You keep up a discussion without losing your temper, even when contradicted	0.564	0.318
You balance time between work and leisure	0.872	0.760
**Subscale 4—Physical activity dimension**	**Regression weight**	**Individual Reliability**
You perform at least 30 min of moderate/intense physical activity, continuously or cumulatively, 5 or more days during the week	0.794	0.630
You do exercises involving muscle strengthening and stretching at least twice a week	0.885	0.783

## Data Availability

The data that support the findings of this study are available from the corresponding author upon reasonable request.

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
