# Peer review of "Adaptation and Validation of the Individual Lifestyle Profile Scale of Portuguese Older Adults Living at Home"

_ijerph, 2022, doi:10.3390/ijerph19095435_

Round 1

Reviewer 1 Report

This study is interesting and clinically useful to comprehend the lifestyle of the older adult population. Also, the study is an important contribution to assessing the lifestyle of older people.

The authors described well the methods of factor analysis and the significance of each individual factor. Despite that, some corrections are needed to improve the article.

Move this sentence (in line 289) in methods or in limitation if the authors think that it is a some lack of questionnaires:   “Since we wrote the original instrument in Brazilian Portuguese we did not translate the scale, however, we made semantic adaptations, especially in what concerns the construction of the sentences and replacement of some terms that we do not use in European Portuguese.” Also after sentence in line 288, authors should be given the explanation for this assessment.

In this article it is missing to explain the limitations and strength of this study. The authors should consider trying to describe limitations in this study. This is very important to add at the end of the article. The only limitation is mentioned in the conclusion. However, the limitation should not be in the conclusion.  For example, line 406 and 407: "...thus considering the influence that contexts have on people's lifestyles this fact is a limitation to our study". Please take out this part of the sentence from the conclusion. The conclusion should allow for possible future implications as well as the importance that the questionnaire may have for the well-being of older people’s lifestyle and causes of risk for the loss of years of healthy life and this should be emphasized.

Reviewer 2 Report

Note:  The authors referenced Pender's Health Promotion Model several times, using the pronoun "he" - Nola Pender, author of Pender's Health Promotion Model, is female.  

Well designed; clearly reported.

Reviewer 3 Report

Dear authors

Congratulations for the research “Adaptation and validation of the Individual Life style Profile 2 Scale of the Portuguese elderly living at home”.

Title:  Adequate. Title:  Actually, the use of the word “elderly” is no longer recommend. Please adopted the “older adults” “older person” in the title and revised all the manuscript.

Abstract: Results need data and be more clear. Conclusion need improve

Introduction: Adequate. However, the author must highlight with strong evidence the relationship between life style and health ageing. Many reinforce with social determinants of health and the framework work of the health ageing (intrinsic capacity- Functional ability)

Methods.  

Globally, the methods need be better organized: sample, instruments, data collection, data analysis, ethical considerations?

No information about the first steps of validation – translations/ adaptation. Content validity +++

The authors performed a EFA and CFA with the same sample? Slip-half the sample? There is no recommendation to used the same sample to performed the EFA and CFA.

“preserved cognitive ability ascertained at the beginning of the interview, supported by the following items: orientation and memory of theMini-Mental State Examination Scale (MMSE) [20].”  What item of memory – registration, recall?

We based the definition of the number of participants on the COnsensus-based 86 Standards for the selection of health Measurement INstruments (COSMIN) recommenda- 87 tion, which suggests six to ten observations for each item of the instrument that need to 88 be validated [21]  -what was the criteria in this study? Be objective

The sample is not Cleary and a bit confused

Clarify, from the 2300 you obtained 300? Provide information about this process.

“40 people refused to participate.”  - lost in the text. Add more information.

More information about the validation of Individual Lifestyle Profile’ Scale – original and other validations study ….Portuguese translations and adaptations?

No information about the sociodemographic items in the instrument.  

Please provide the fitting index. “In the CFA the quality of fit of the model was assessed, namely through the calcula- 122 tion of quality of fit coefficients such as the Comparative Fit Index (CFI), the Goodness- 123 of-Fit Index (GFI), the Adjusted Goodness-of-Fit Index (AGFI), the Incremental Fit Index 124 (IFI), the Tacker-Lewis Index (TLI), the Root Mean Square Residual (RMSR), the Root 125 Mean Square Error of Approximation (RMSEA) and the Modified Expected Cross-Vali- 126 dation Index (MECVI).”

Star with this information

“Considering that this study is integrated in a larger research project entitled: ‘Frail elderly at home: Sensitive gains of rehabilitation nursing care’, we requested and obtained permission from the authors of the instrument to use and validate the ILP.

Results.  

Review the tables – information in PT

Need be organized. To must table (e.g. KMO). Add and Bartlett's Test of Sphericity

“Goodness-of-Fit Index (GFI) and the Adjusted  Goodness-of-Fit Index (AGFI) are 0.893 and 0.848 respectively, and therefore we can consider that these values indicate a good fit. – according Marroco is a poor adjustment

Discussion. Repeat many results.

The limitation of the study was also be discussed, but need a deep analysis, specially related with data collection.

Organization and style of presentation: OK

In the manuscript the citation and reference must be review and a carefully edition will be relevant.

Good luck and thank you very much for giving me a chance to review.

Round 2

Reviewer 3 Report

Dears authors 

Thank you for the revision of the paper. 

The methods section is more clear and more organized.

The text needs editions 

Best regards